# Workplace Bullying and Long-Term Sickness Absence—A Five-Year Follow-Up Study of 2476 Employees Aged 31 to 60 Years in Germany

**DOI:** 10.3390/ijerph19127193

**Published:** 2022-06-11

**Authors:** Hermann Burr, Cristian Balducci, Paul Maurice Conway, Uwe Rose

**Affiliations:** 1Federal Institute for Occupational Safety and Health (BAuA), Nöldnerstraße 40-42, 10317 Berlin, Germany; rose.uwe@baua.bund.de; 2Department of Psychology, University of Bologna, Viale Berti Pichat, 5, 40127 Bologna, Italy; cristian.balducci3@unibo.it; 3Department of Psychology, University of Copenhagen, Øster Farimagsgade 2A, 1165 Copenhagen, Denmark; paul.conway@psy.ku.dk

**Keywords:** long-term sickness absence, sickness absence, bullying, harassment, prospective analyses

## Abstract

Objectives: The aim was to investigate workplace bullying as a risk factor for five-year occurrence of long-term sickness absence (LTSA) in a representative cohort of employees in Germany. Methods: In the German Study on Mental Health at Work (S-MGA) (*n* = 2476), episodes of long-term sickness absence (LTSA) between baseline and follow-up were assessed in the follow-up interview. Workplace bullying was measured in the baseline interview using a hybrid approach, which combines the behavioural experience and self-labelling methods. Through binomial regressions, the association of baseline level of workplace bullying with first-episode LTSA during follow-up was estimated, adjusting for baseline age, gender, occupational level, smoking status and physical demands at work. Results: Severe bullying heightened the risk for LTSA by approximately 50% (Rate ratio—RR: 1.48, 95% Confidence interval—CI: 1.05; 2.19), while occasional bullying heightened the risk by 15% (RR: 1.15, CI: 0.85; 1.55). When excluding LTSA events occurring in the first 2 years, the associations between bullying and LTSA increased by approximately one third. Conclusions: Workplace bullying seems to be a risk factor for LTSA even when controlling for occupational level, smoking and physical demands at work and when taking possible reverse causality into account. We suggest to investigate effects of LTSA in more settings, to distinguish between occasional and severe bullying and employ longer follow-up intervals.

## 1. Introduction

Workplace bullying can be defined as systematic and persistent exposure to negative acts at work such as verbal mistreatment and abuse, social isolation, and withholding of information that affects performance. To apply the label bullying to such negative behaviours, they have to occur frequently over a period of time [1]. Exposure to bullying may in the long run lead to severe consequences for mental health [2].

One outcome of workplace bullying that has received consistent attention for its significant consequences for individual and organizations is sickness absence [3]. Exposure to bullying might affect sickness absence through a health pathway [4]. Bullying might lead to long-term stress reactions, which can, among other consequences, result in depressive symptoms [5,6,7,8,9,10,11,12,13,14]. It has been repeatedly shown that poor mental health increases the risk of sickness absence [15,16,17,18,19]. Based on this evidence, workplace bulling is thus expected to increase the risk of sickness absence.

A review of prospective studies on workplace bullying and sickness absence (Nielsen et al., 2016) and three recent prospective investigations [20,21,22] indicate that workplace bullying is a risk factor for sickness absence. However, all of these studies were carried out in the Nordic countries, except one that was conducted in Belgium [23]. This Belgian study and four Nordic studies examined long-term sickness absence (LTSA) spells using cut-off points ranging from about 2 to 8 weeks [21,22,23,24,25]; two studies included shorter spells [26,27], while the rest investigated total sickness absence, that is, a summary measure of sickness absence days or weeks. Most studies of LTSA included follow-ups of 1.5 years or less; only one study employed a longer follow-up of 7.3 years [24].

Building on previous research, the aim of the present study is to further contribute to shed light on the relationship between workplace bullying and sickness absence. Specifically, the study sought to investigate workplace bullying as a risk factor for long-term sickness absence (LTSA) in a representative cohort of employees in Germany who were followed up for five years. Some types of consequences of workplace bullying may take time to emerge—especially so in the case of long-term sickness absence that may indicate serious health consequences resulting from the exposure to workplace bullying. Therefore, an extended period of time might be necessary for workplace bullying to exert its true effect on long-term sickness absence.

## 2. Materials and Methods

### 2.1. Sample

We used data from the German Study on Mental Health at Work (S-MGA), which is a nation-wide representative employee cohort study with a baseline survey in 2012 and a follow-up survey in 2017 [28]. At baseline, the target population consisted of all individuals employed in Germany as of 31 December 2010, born in 1951–1980 [28]. The study population was enrolled through the register of Integrated Employment Biographies (IEB) of the German Federal Employment Agency at the Institute for Employment Research (IAB). This register covers all employees subject to social security contributions, thus excluding civil servants, self-employed workers and freelancers. The analysed cohort included 2476 persons that were employed at baseline (Figure 1). In the main analysis, participants were followed up between 2012 and 2017 for their first episode of LTSA.

In the analysed cohort, men and women were equally represented, mean age was almost 47 years, the most prevalent occupational level group consisted of skilled workers, non-smokers were more prevalent than former or current smokers were, and physical demands were relatively low (Table 1). Most participants reported that they have been never bullied; 25% (*n* = 613) experienced an episode of long-term sickness absence (LTSA) during the 5-year follow-up.

### 2.2. Measures

All information used for the present study was obtained through interviews in the respondents’ home [28].

#### 2.2.1. Long-Term Sickness Absence (LTSA)

As dependent variable, we focused on self-reported episodes of LTSA between baseline (2012) and follow-up (2017). LTSA was defined as having reported at least one episode of LTSA lasting ≥6 weeks, which according to the German law is the minimum duration of a LTSA episode for it to be compensated by the sickness insurance [29]. Information on LTSA was based on a question at follow-up (2017) regarding experienced LTSA episodes since baseline (2012) [30] (“Have you been ill for a longer period since last interview, i.e., at least 6 weeks or in a medical rehab? Please name each event, even if it has ended, as a separate spell.” followed by the question “From when to when were you sick for a long period of time or in medical rehabilitation? Please tell me the beginning and end month and year.”).

#### 2.2.2. Workplace Bullying

Workplace bullying was assessed with the following two questions: “Do you frequently feel unjustly criticized, hassled or shown up in front of others by co-workers?” and “Do you frequently feel unjustly criticized, hassled or shown up in front of others by superiors?”, to be answered using the response options “yes” or “no”. Each of these two questions was followed by the item: “And how often did it occur in the last 6 months?”, to be answered using the following response options: “daily”, “at least once a week”, “at least once a month” and “less than once a month”. This hybrid approach combines the behavioural experience and the self-labelling methods [31] and showed the same predictive validity as the reporting of negative acts based on the behavioural experience method alone [32]. We distinguished between severe bullying (i.e., being bullied at least weekly by supervisors, colleagues or both), occasional bullying (i.e., being bullied at least sometime within the last 6 months by supervisors, colleagues or both) and no bullying [33].

#### 2.2.3. Covariates

We considered gender, age, occupational level, smoking and physical demands at work as potential covariates and these factors have been previously associated to LTSA [31,34,35,36]. In the present data, chi-square tests in cross-tabulations showed that gender and age were not significantly associated with bullying (*p* = 0.484; *p* = 0.501). Bullied workers had a lower occupational level (*p* = 0.001) and higher physical demands at work (*p* < 0.000), and were more likely to be smokers (*p* = 0.010) (Tables not shown).

Information on gender and age was collected through the interviews. Occupational level was categorized into the following four groups according to the International Standard Classification of Education (ISCED), which was based on the International Standard Classification of Occupations (ISCO 08): unskilled workers, skilled workers, semi-professionals, and academics/managers [37]. In contrast to ISCED—which did not classify managers into any educational level, we grouped managers together with academics as in other classifications of socio-economic level [38]).

With regard to smoking status, we distinguished between three groups, namely “never”, “former” and “current” smokers [39].

Physical demands at work were measured with a scale based on three items covering standing posture, awkward body postures and carrying and lifting, adapted from the BiBB/BAuA employment study [40,41]: “How often do you have to: ‘work in a standing position?’, ‘work in a bent, squatted, kneeling, lying or overhead position?’ or ‘carry or lift heavy loads (women > 10 kg, men > 20 kg)?’. The response options were “never” (0), “up to 1/4 of the time” (1), “up to half of the time” (2), “up to 3/4 of the time” (3), “more than three quarters (almost all of the time) (4)”. Cronbach’s alpha for this scale was 0.76, while inter-item correlations ranged from 0.51–0.58.

### 2.3. Analysis

Through a binomial regression analysis allowing for the calculation of rate ratios (RR’s) [42], we estimated associations of baseline level of workplace bullying with first episode of LTSA during follow-up. We adjusted for baseline age, gender, occupational level, smoking status and physical demands at work. Both age and age squared were entered into the model; the latter was included because of its association with LTSA, which was higher from 31 to 50 years and then remained stable between 51 to 60 years.

In a sensitivity analysis, we left out LTSA episodes occurring in the first 2 years of follow-up to reduce the potential impact of reverse causality (i.e., LTSA as a predictor of workplace bullying). LTSA episodes occurring just before baseline could be risk factors for bullying and lead to a higher risk of LTSA episodes close after baseline. Removing cases of LTSA close after baseline is a way of handling this possible bias [43,44].

We note that, instead of a binomial regression, we first considered to perform a cox regression using time to event data; however, we eventually decided not to perform such an analysis because the proportional hazards assumption was violated.

As gender did not interact with other independent variables in the prediction of LTSA, we decided not to stratify for this variable.

The analyses were conducted by means of SPSS 25 (IBM SPSS, Chicago, IL, USA), using the GENLIN command (link = logit) to carry out binomial regressions [42].

## 3. Results

Table 2 shows the results of the binomial regression analysis examining the five-year prospective association between workplace bulling at baseline and LTSA at follow-up. After adjusting for the covariates considered, severe bullying was associated with an elevated risk of LTSA (RR = 1.48; 95% CI = 1.05–2.19). Occasional bullying also heightened the risk of LTSA (RR = 1.15; 95% CI = 0.85–1.55), although the 95% confidence intervals included the unity.

Table 3 shows the results of the sensitivity analysis that we carried out to minimize the effects of reverse causality by excluding LTSA events occurring in the first 2 years. This analysis showed stronger associations between workplace bullying and LTSA, with point estimates increased by approximately one third for both severe and occasional bullying.

## 4. Discussion

The results of the present study indicate that workplace bullying is a risk factor for subsequent LTSA. Such an association remained also when controlling for covariates such as occupational level, smoking and physical demands at work or when taking possible reverse causality into account. The effect of workplace bullying was stronger for LTSA episodes that took place ≥ 2 years after baseline. This highlights the importance of employing long follow-up intervals when predicting outcomes of workplace bullying such as LTSA.

These findings are compatible with the view that bullying can be an escalating process [33,45] whose extreme consequences may take time to develop (e.g., severe health problems resulting in an individual staying away from work for a long period of time). Supporting this, a previous qualitative study [46] showed that, in the initial stages of bullying victimization, targets try to employ a variety of coping initiatives, such as constructive conflict solving strategies and fighting back. Only later, when these attempts to resolve the situation prove to be unsuccessful, victims may develop a sense of powerlessness leading to compromised health and LTSA as a possible consequence. In particular, the association between workplace bullying and LTSA may be mediated by depressive symptoms. Bullying has been found to be a risk factor for depressive symptoms [5,6,7,8,9,10,11,12,13,14], which is turn might result in LTSA [15,16,17,18,19]. The fact that in our study the association between workplace bullying and LTSA increases over time, supports the hypothesis that depressive symptoms precede the occurrence of LTSA. There might also be other health related factors being mediators of the association between bullying and LTSA, such as emotional exhaustion, anxiety, sleeping problems and chronic somatic diseases [47,48,49,50]. An alternative or complementary explanation could be that LTSA is a coping strategy that that victims adopt in the long run to avoid the context in which bullying started and escalated [51].

### 4.1. Strengths and Limitations

A first strength of the present study is its prospective design, which increases the possibilities to draw causal conclusions [52]. In addition, the relatively long follow-up we employed allowed us to take reverse causality into account [43,44]. Second, the prospective design reduced same method bias as data on workplace bullying were collected at the baseline interview, whereas data on LTSA between baseline and follow-up were collected at the follow-up interview. Third, the study did not comprise workers older than 60 years at baseline, which may have limited selection out of employment through unemployment or pensioning due to sickness absence [53].

The strengths of this study need to be balanced against its weaknesses. One is that a low response rate could have biased the results. However, based on a comparison between our sample and the population from which it was drawn, the bias due to regional characteristics, gender, age, education, profession and income was limited [28,54]. A second limitation is that the study does not cover employees younger than 30 years or those being civil servants or self-employed. Bullying may occur more often among younger workers according to previous evidence showing that it is more prevalent among employees in their 30′ies than among employees in their 50′ies [31]. As we are not able to assess if bullying is more or less prevalent among civil servants and self-employed, the potential impact that the lack of inclusion of these employees has on the present findings is unknown. A third limitation is that we do not have information regarding exposure to workplace bullying between baseline and follow-up, which would have supported a sounder understanding of the causal process linking bullying and sickness absence [55].

### 4.2. Comparison with Earlier Studies

To our knowledge, four prospective studies have hitherto examined workplace bullying as risk factor for LTSA. In these studies, LTSA lasted 2 to 8 weeks, while follow-up intervals ranged from 1 to 7.3 years [21,23,24,25]. The associations we found in our study were similar to those observed in the only study that included a relatively long follow-up (mean 7.3 years) and considered also shorter LTSA spells (i.e., ≥2 weeks), which found a risk for overall workplace bullying of 1.28 that lies between our risk estimates of 1.15 and 1.48 [24]. Only one previous study, which also considered short-term sickness absence, distinguished between occasional and severe bullying and found, in line with our study, that severe bullying had a stronger effect on LTSA than occasional bullying [25]. The results of the other two studies, which did not distinguish between occasional and severe bullying, found risk estimates of 1.3 to 1.6 for LTSA lasting 2–3 weeks [21,23]. As these studies included shorter episodes of LTSA, their results are not comparable with those obtained in our study. In conclusion, there is a paucity of studies on workplace bullying and LTSA allowing us to place the findings of the present study in a broader international context.

### 4.3. Conclusions

The present study suggests that a reduction in workplace bullying could lead to a decrement in episodes of LTSA. As psychosocial factors seem to have an important effect on future cases of bullying [56,57,58], an improvement of the psychosocial working environment might contribute to a reduction of bullying. An improved psychosocial safety climate and more support of victims might also be instrumental in alleviating the health consequences of bullying, thus contributing to the prevention of LTSA [59,60,61,62].

Given the possible role of workplace bullying as risk factor for LTSA, it is recommended to further investigate this association in several occupational settings and countries, to look at episodes of sickness absence of longer duration and to employ longer follow-up periods, as well as to distinguish between occasional and severe bullying.

## Figures and Tables

**Figure 1 ijerph-19-07193-f001:**
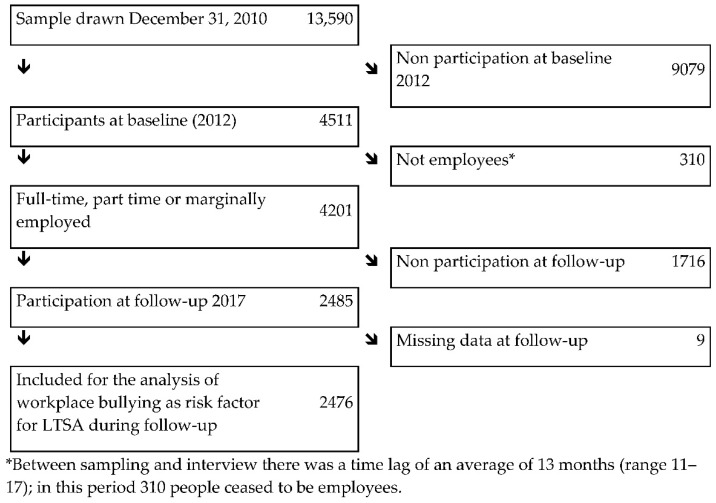
Flow diagram of participation in SMGA’s 2012 baseline and the 2012–2017 cohort.

**Table 1 ijerph-19-07193-t001:** Distribution of independent and depent variables among among 2476 employees aged 31 to 60 years in Germany in 2012.

	*n*	%	Mean	Std. Dev.
GENDER 2012				
Men	1206	49%		
Women	1270	51%		
AGE 2012			46.9	7.6
OCCUPATIONAL LEVEL 2012				
Unskilled workers	144	6%		
Skilled workers	1039	42%		
Semi-professionals	683	28%		
Academics/managers	610	25%		
SMOKING STATUS 2012				
Never	1009	41%		
Former	765	31%		
Current	702	28%		
PHYSICAL DEMANDS 2012 (min 0, max 4)			1.09	1.02
WORKPLACE BULLYING 2012				
No	2069	84%		
Occasional	244	10%		
Severe	163	7%		
LTSA EPISODE 2012–2017				
No	1863	75%		
Yes	613	25%		

LTSA: Long term sickness absence ≥ 6 weeks.

**Table 2 ijerph-19-07193-t002:** Associations between baseline workplace bullying in 2012 and 613 cases of long term sickness absence (LTSA) 2012–2017 among 2476 employees aged 31 to 60 years in Germany in 2012. Multiple binomial regressions. Rate Ratios (RR).

	N	LTSA during Follow-Up 2012–2017				
		Cases, *n*	Cases (%)	Adjusted for Gender, Age, Age Squared, Occupational Level, Smoking and Physical Demands at Work		
				*p*-Value *	RR	95% CI
Workplace bullying 2012				0.072		
no	2069	489	24		1	
occasional	244	68	28		1.15	0.85; 1.55
severe	163	56	34		1.48	1.05; 2.19

LTSA: Long-term sickness absence ≥ 6 weeks. * This *p*-value denotes to what extent the entire categorical workplace bullying variable was associated to LTSA.

**Table 3 ijerph-19-07193-t003:** Associations between baseline workplace bullying in 2012 and 405 cases of long term sickness absence (LTSA) 2014–2017 * among 2476 employees aged 31 to 60 years in Germany in 2012. Multiple binomial regressions. Rate Ratios (RR).

	N	LTSA during Follow-Up 2014–2017 ^*^				
		Cases, *n*	Cases (%)	Adjusted for Gender, Age, Age Squared, Occupational Level, Smoking and Physical Demands at Work		
				*p*-Value ^†^	RR	95% CI
Workplace bullying 2012				0.005		
no	2069	317	15		1	
occasional	244	47	19		1.21	0.86; 1.71
severe	163	41	25		1.69	1.10; 2.36

LTSA: Long-term sickness absence ≥ 6 weeks. * Cases of LTSA occurring <2 years after baseline exposure of workplace bullying were excluded. ^†^ This *p*-value denotes to what extent the entire categorical workplace bullying variable was associated to LTSA.

## Data Availability

A scientific use file (SUF) containing both wave 1 and wave 2 of the cohort is available at the Research Data Centre of the Federal Institute of Occupational Safety and Health.

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
