# Peer review of "Workplace Bullying and Long-Term Sickness Absence—A Five-Year Follow-Up Study of 2476 Employees Aged 31 to 60 Years in Germany"

_ijerph, 2022, doi:10.3390/ijerph19127193_

Round 1

Reviewer 1 Report

 The manuscript addresses an important topic for public health and occupational health, which is the association between workplace bullying and long-term sickness absence. The authors used data from a representative national health survey in Germany. The study is well justified as most of the studies on the theme are from Nordic countries, and previous studies evaluated shorter periods of follow-up. The results are interesting and the methods are well applied, in general. The discussion is relevant, although some more limitations may be discussed. A few comments are made below for a few clarifications and possible improvement.

In the introduction, the authors state a correct definition of bullying, from Einarsen et al, 2020. However, they mentioned that it would only be defined as workplace bullying when the negative acts occur at least about six months. This is not exactly the definition stated by Einarsen and other authors. It is a consensus that bullying is a type of violence that occurs repeatedly and over a period of time, but six months is only a general time reference. It is not an obligation to last at least six months. I would suggest to revise that. Also, in the end of the introduction, the authors mentioned that Behavioural consequences of workplace bullying may take time to emerge especially so in the case of long-term sickness absence that may indicate serious health consequences resulting from the exposure to workplace bullying. Therefore, an extended period of time might be necessary for workplace bullying to exert its true effect on long-term sickness absence. I found this excerpt confusing. What do the authors mean when they cite behavioural consequences of bullying. I would not consider sickness absence as a behavioural consequence. I would suggest to the authors to revise or clarify this aspect.

Regarding the methods, I believe there are some parts that are actually results, not methods. For example, the part of the text where the authors describe that: In the analysed cohort, men and women were equally represented, mean age was almost 47 years, the most prevalent occupational level group consisted of skilled workers, non-smokers were more prevalent than former or current smokers were, and physical demands were relatively low (Table 1). Most participants reported that they have been never bullied; 25% (n=613) experienced an episode of long-term sickness ab-sence (LTSA) during the 5-year follow-up. I would suggest to place this part of the text in the results section. In the methods section, it is necessary just a good description of how participants were selected, describing also all procedures that were adopted.

Concerning the measures, the authors used only two questions to evaluate the main exposure, workplace bullying. Were those questions previously validated? Are they enough to evaluate the concept of workplace bullying? Why? Any reason for not using validated instruments and/or a direct question with a previous definition of bullying, as suggested and used by previous publications? I believe this should be clarified and may need some additional discussion of limitations in the following section. Concerning the methods to measure the outcome, it is not totally clear in the text whether the outcome was evaluated only by self-report or there was other way used for that. This should be clarified and may also be discussed as a limitation.

Regarding the covariates, any reason for not including other covariates and possible confounders in the analyses? As a national survey, there might have much more valuable information available to be used. Educational level, socioeconomic status, other behavioural and individual factors, as well as other occupational characteristics may also play a role on the association between bullying and LTSA. If possible, it would be interesting to include these variables in the analyses. If not, more limitations may be discussed.

Concerning the discussion, the authors state that extreme consequences take time to develop. However, even though bullying may occur as an escalating process, serious consequences may take place within short periods of experiencing bullying. Surely a longer exposure (sometimes years) may be associated with worse health outcomes and more sickness absence. Also, a longer period of follow-up may be able to demonstrate other extreme consequences. Nonetheless, it is not totally true that those consequences take time to develop, as many serious consequences may occur in short periods.

The authors discussed an important pathway between bullying and LTSA trough depression. However, this was the only mediator cited and discussed by the authors. On the other hand, workplace bullying has been associated with several other health outcomes, such as anxiety, sleeping problems and other stress-related diseases, chronic diseases, musculoskeletal pain, etc. If the authors believe those other outcomes may be important for the association between bullying and LTSA, some additional discussion on this topic would reinforce this part of the discussion.

Concerning the limitations, I do not agree that the observational design is a limitation by itself. As mentioned by the authors, it would not be possible to design an experimental study to assess this research question. Most epidemiological studies are observational. The most important to assure quality is to implement the best methods available for each case.

Finally, regarding the conclusions and recommendations, the authors only mentioned that a reduction in workplace bullying could lead to a decrement in episodes of LTSA, also recommending further research in other work settings and countries. This is fine, but were not other recommendations and conclusions raised by the results? I believe it would be important to explore a deeper discussion in this part, including other aspects that may be important for policies and for scientific research on the subject. What should be recommend for everything that happens between a new case of bullying and the occurrence of sickness absence?

Minor comments:

Please revise p value = 0.000 in page 5. P=0.000 dos not exist, so it would be better to use p<0.001.

Please, revise the sentence In particular, workplace bullying may lead to LTSA due to the established casual role that the former has in the development of depressive symptoms!". It sounds confusing. Revise also the term casual.

Page 8, item 4.1: revise the term casual conclusions.

Author Response

Reviewer 1

GENERAL COMMENT FROM AUTHORS. All changes in the 1st revision of the manuscript are highlighted with yellow background. Importantly, we apologize regarding page numbers in the submitted revision. Page numbers in our responses to the reviewers below refer to the true number of pages in the file, not by the numbers given on the top page, which occurs from the third page in the manuscript. We were not able to correct this inconsistency. We have added line numbers in the manuscript which are not hampered by this inconsistency.

REVIEWERS NUMBERED COMMENTS AND SUGGESTIONS:

1.1. The manuscript addresses an important topic for public health and occupational health, which is the association between workplace bullying and long-term sickness absence. The authors used data from a representative national health survey in Germany. The study is well justified as most of the studies on the theme are from Nordic countries, and previous studies evaluated shorter periods of follow-up. The results are interesting and the methods are well applied, in general. The discussion is relevant, although some more limitations may be discussed. A few comments are made below for a few clarifications and possible improvement.

AUTHORS RESPONSE: Thank you for the comment.

1.2. In the introduction, the authors state a correct definition of bullying, from Einarsen et al, 2020. However, they mentioned that it would only be defined as workplace bullying when the negative acts occur “at least about six months”. This is not exactly the definition stated by Einarsen and other authors. It is a consensus that bullying is a type of violence that occurs repeatedly and over a period of time, but six months is only a general time reference. It is not an obligation to last at least six months. I would suggest to revise that. Also, in the end of the introduction, the authors mentioned that “Behavioural consequences of workplace bullying may take time to emerge – especially so in the case of long-term sickness absence that may indicate serious health consequences resulting from the exposure to workplace bullying. Therefore, an extended period of time might be necessary for workplace bullying to exert its true effect on long-term sickness absence”. I found this excerpt confusing. What do the authors mean when they cite “behavioural consequences” of bullying. I would not consider sickness absence as a “behavioural consequence”. I would suggest to the authors to revise or clarify this aspect.

AUTHORS RESPONSE:

(Page 1, lines 42-43): Yes, there is a difference between the theoretical definition and practical applications of such a definition. We now abstain from referring to specific numbers regarding frequency and length of bullying acts in the first paragraph of the introduction on page 1.

(Page 2, line 72): Yes, there are different views on what can be termed behaviour .We now write ‘Some types of consequences’ instead of ‘behavioural consequences’.

1.3. Regarding the methods, I believe there are some parts that are actually results, not methods. For example, the part of the text where the authors describe that: “In the analysed cohort, men and women were equally represented, mean age was almost 47 years, the most prevalent occupational level group consisted of skilled workers, non-smokers were more prevalent than former or current smokers were, and physical demands were relatively low (Table 1). Most participants reported that they have been never bullied; 25% (n=613) experienced an episode of long-term sickness ab-sence (LTSA) during the 5-year follow-up.” I would suggest to place this part of the text in the results section. In the methods section, it is necessary just a good description of how participants were selected, describing also all procedures that were adopted.

AUTHORS RESPONSE: We are aware that there are two traditions regarding numerical description of the population and variables used to answer the research question of a manuscript. One tradition includes this in the beginning of the results section. Another tradition sees such descriptions of the population and variables as a part of the method section. According to this tradition the results section is only occupied with answering the research question. We follow this other tradition.

 1.4. Concerning the measures, the authors used only two questions to evaluate the main exposure, workplace bullying. Were those questions previously validated? Are they enough to evaluate the concept of workplace bullying? Why? Any reason for not using validated instruments and/or a direct question with a previous definition of bullying, as suggested and used by previous publications? I believe this should be clarified and may need some additional discussion of limitations in the following section. Concerning the methods to measure the outcome, it is not totally clear in the text whether the outcome was evaluated only by self-report or there was other way used for that. This should be clarified and may also be discussed as a limitation.

AUTHORS RESPONSE: Thank you for these clarifying questions on the measurement of bullying and long term sickness absence.

Bullying (Page 4, line 127) : This measurement of bullying was validated by Garthus Niegel (2016). In reference 32 this validation study is described in detail. In the 1st version of the submitted manuscript we wrote: ‘”. This hybrid approach combines the behavioural experience and the self-labelling methods [31] and showed the same predictive validity as the reporting of negative acts based on the behavioural experience method alone [32].’

Long term sickness absence (Page 3, line 103-114): This way of asking about sickness absence spells was developed inspired by an employment history tool developed and validated in the SHARE Study by Börsch-Supan et al 2013, see reference 30. We now add in more detail how we asked about sickness absence spells in the end of the paragraph (Page 3, line 110-114).

 1.5. Regarding the covariates, any reason for not including other covariates and possible confounders in the analyses? As a national survey, there might have much more valuable information available to be used. Educational level, socioeconomic status, other behavioural and individual factors, as well as other occupational characteristics may also play a role on the association between bullying and LTSA. If possible, it would be interesting to include these variables in the analyses. If not, more limitations may be discussed.

AUTHORS RESPONSE (page 4, line 134-Page 5, line 161): As already indicated in the manuscript, we included the following covariates in the analyses: Gender, age, occupational level, smoking and physical demands at work as these factors have been previously associated to LTSA. We have already given a number of references showing why these factors were selected, refs 34-37. In order to avoid statistically unstable models with small – or empty – cells, we abstained from controlling for other factors than those mentioned. We included occupational level, which is a good indicator of both educational level, socioeconomic status, less favourable working conditions and a number of lifestyle factors being risk factors for LTSA. We therefore included only one powerful working condition as risk factor for LTSA, namely physical demands at work, and only one powerful lifestyle risk factor for LTSA, namely smoking.

 1.6. Concerning the discussion, the authors state that “extreme consequences take time to develop”. However, even though bullying may occur as an escalating process, serious consequences may take place within short periods of experiencing bullying. Surely a longer exposure (sometimes years) may be associated with worse health outcomes and more sickness absence. Also, a longer period of follow-up may be able to demonstrate other extreme consequences. Nonetheless, it is not totally true that those consequences take time to develop, as many serious consequences may occur in short periods.

AUTHORS RESPONSE (Page 6, line 222-Page 7, line 225): Thank you for this comment. We have now added a ‘can be’ and a ‘may’ in the sentence: ‘These findings are compatible with the view that bullying can be an escalating process [33, 46] whose extreme consequences may take time to develop (e.g., severe health problems resulting in an individual staying away from work for a long period of time).’

1.7. The authors discussed an important pathway between bullying and LTSA trough depression. However, this was the only “mediator” cited and discussed by the authors. On the other hand, workplace bullying has been associated with several other health outcomes, such as anxiety, sleeping problems and other stress-related diseases, chronic diseases, musculoskeletal pain, etc. If the authors believe those other outcomes may be important for the association between bullying and LTSA, some additional discussion on this topic would reinforce this part of the discussion.

AUTHORS RESPONSE (Page 7, line 237-240): Thank you for this comment. We have now added the sentence: “There might also be other health related factors being mediators of the association between bullying and LTSA, such as emotional exhaustion, anxiety, sleeping problems and chronic somatic diseases [48-51]“

1.8. Concerning the limitations, I do not agree that the observational design is a “limitation” by itself. As mentioned by the authors, it would not be possible to design an experimental study to assess this research question. Most epidemiological studies are observational. The most important to assure quality is to implement the best methods available for each case.

AUTHORS RESPONSE (Page 7, line 255-258): You are right. We now simply write ‘One is that a low response rate could have biased the results. However, based on a comparison between our sample and the population from which it was drawn, the bias due to regional characteristics, gender, age, education, profession and income was limited [28, 55]’ instead of a much lengthier wording referring to ‘observational studies’.

1.9. Finally, regarding the conclusions and recommendations, the authors only mentioned that “a reduction in workplace bullying could lead to a decrement in episodes of LTSA”, also recommending further research in other work settings and countries. This is fine, but were not other recommendations and conclusions raised by the results? I believe it would be important to explore a deeper discussion in this part, including other aspects that may be important for policies and for scientific research on the subject. What should be recommend for everything that happens between a new case of bullying and the occurrence of sickness absence?

AUTHORS RESPONSE (Page 8, lines 292-298):  Good point. We have now added the sentences: ‘As psychosocial factors have an important effect on future cases of bullying [57-59], an improvement of the psychosocial working environment might contribute to a reduction of bullying. An improved psychosocial safety climate and more support of victims might also be instrumental in alleviating the health consequences of bullying, thus contributing to the prevention of LTSA [60-63]’

1.10. Minor comments:

Please revise p value = 0.000 in page 5. P=0.000 dos not exist, so it would be better to use p<0.001.

AUTHORS RESPONSE (Page 4, line 141): We corrected this.

1.11. Please, revise the sentence “In particular, workplace bullying may lead to LTSA due to the established casual role that the former has in the development of depressive symptoms!". It sounds confusing. Revise also the term “casual”.

AUTHORS RESPONSE:

(Page 7, lines 231-234): We now write two sentences ‘In particular, the association between workplace bullying and LTSA may be mediated by depressive symptoms. Bullying has been found to be a risk factor for depressive symptoms [5-14], which is turn might result in LTSA [15-19].’ instead of the previously wording.1.12. Page 8, item 4.1: revise the term “casual conclusions”.

AUTHORS RESPONSE (Page 7, line 245): We now write ‘causal’.

Reviewer 2 Report

The paper covers an interesting and up-to-date topic. The article presents the evaluation of association between bullying at work and long-term sickness absence. The analysis concerns Germany and is based on data from a nationwide survey. The strength of the work is that a representative cohort of employees is taken into account, which allows the authors to draw general conclusions. Another advantage is that the authors consider a relatively long time of observation, i.e. five year follow-up, which is not common in other studies. Longer period of time may contribute to better understanding of the phenomenon and assessment of the influence of workplace bullying on long-term sickness absence. Adequate statistical method was used – binominal regression with gender, age, occupational level, smoking and physical demands as covariates. The results show that workplace bullying can be considered as a risk factor for long-term sickness absence. The paper is a valuable contribution in this research area.

However, I have dome detailed remarks:

1.  It is not common to give the sample size in the title of article. I recommend considering removing the number “2,476” from the title.

2.  Although the cohort is representative some working persons are excluded from the register based on which the sample was drawn (as you mention: civil servants, self-employed workers and freelancers). I find it as a kind of limitation of the study, particularly in case of civil servants as the other two groups may be less affected by bullying. Consider if you also would recognize it as a limitation and if so include it in the section Discussion in Strengths and limitations.

3.       At the end of p.4 and at the beginning of p.5 you write:

“In the present data, cross-tabulations showed that gender and age were not significantly associated with bullying (p = 0.484; p = 0.501). Bullied workers had a lower occupational level (p = 0.001), higher physical demands at work (p = 0.000) and were more likely to be smokers (p =0.010) (Tables not shown).”

You show p-values but it is not clear to the reader what statistical test(s) were used. As the cross-tabulation is mentioned in the first sentence one may suppose that chi-square test was used to check the association between bullying and gender and age. Please clarify what statistical procedure was used to evaluate relationships described in this fragment.

  1. Finally, a remark about citing (references). The journal uses a different way of giving references than this used in the manuscript, i.e. with consecutive numbers and not authors’ names and year of publication. According to IJERPH Instructions for Authors: “References must be numbered in order of appearance in the text (including table captions and figure legends) and listed individually at the end of the manuscript”.

Author Response

Reviewer 2

GENERAL COMMENT FROM AUTHORS. All changes in the 1st revision of the manuscript are highlighted with yellow background. Importantly, we apologize regarding page numbers in the submitted revision. Page numbers in our responses to the reviewers below refer to the true number of pages in the file, not by the numbers given on the top page, which occurs from the third page in the manuscript. We were not able to correct this inconsistency. We have added line numbers in the manuscript which are not hampered by this inconsistency.

REVIEWERS NUMBERED COMMENTS AND SUGGESTIONS:

2.0. The paper covers an interesting and up-to-date topic. The article presents the evaluation of association between bullying at work and long-term sickness absence. The analysis concerns Germany and is based on data from a nationwide survey. The strength of the work is that a representative cohort of employees is taken into account, which allows the authors to draw general conclusions. Another advantage is that the authors consider a relatively long time of observation, i.e. five year follow-up, which is not common in other studies. Longer period of time may contribute to better understanding of the phenomenon and assessment of the influence of workplace bullying on long-term sickness absence. Adequate statistical method was used – binominal regression with gender, age, occupational level, smoking and physical demands as covariates. The results show that workplace bullying can be considered as a risk factor for long-term sickness absence. The paper is a valuable contribution in this research area.

AUTHORS RESPONSE: Thank you for these remarks.

2.1. However, I have some detailed remarks: It is not common to give the sample size in the title of article. I recommend considering removing the number “2,476” from the title.

AUTHORS RESPONSE (Page 1, line 3): Within occupational epidemiology we often have to deal with small samples due to financial constraints, so sometimes sizes of analysed populations have been reported in the titles of not only larger meta-analyses (Kivimaki et al. 2015) but also of individual studies (Lund et al. 2006). So we suggest to keep the N in the title of the manuscript, if the journal accepts this.

References

Kivimaki M, et al. (2015) Long working hours and risk of coronary heart disease and stroke: a systematic review and meta-analysis of published and unpublished data for 603,838 individuals. Lancet (London, England) 386(10005):1739-46 doi:10.1016/S0140-6736(15)60295-1

Lund T, Labriola M, Christensen KB, Bultmann U, Villadsen E (2006) Physical work environment risk factors for long term sickness absence: prospective findings among a cohort of 5357 employees in Denmark. BMJ 332(7539):449-52 doi:10.1136/bmj.38731.622975.3A

2.2.  Although the cohort is representative some working persons are excluded from the register based on which the sample was drawn (as you mention: civil servants, self-employed workers and freelancers). I find it as a kind of limitation of the study, particularly in case of civil servants as the other two groups may be less affected by bullying. Consider if you also would recognize it as a limitation and if so include it in the section Discussion in Strengths and limitations.

AUTHORS RESPONSE (Page 7, lines 258-266): Thank you for this comment. We have now added the following sentence under limitations: “A second limitation is that the study does not cover employees younger than 30 years or those being civil servants or self-employed. Bullying may occur more often among younger workers according to previous evidence showing that it is more prevalent among employees in their 30’ies than among employees in their 50’ies [37]. As we are not able to assess if bullying is more or less prevalent among civil servants and self-employed, the potential impact that the lack of inclusion of these employees has on the present findings is unknown.”

2.3.       At the end of p.4 and at the beginning of p.5 you write:

 “In the present data, cross-tabulations showed that gender and age were not significantly associated with bullying (p = 0.484; p = 0.501). Bullied workers had a lower occupational level (p = 0.001), higher physical demands at work (p = 0.000) and were more likely to be smokers (p =0.010) (Tables not shown).”

You show p-values but it is not clear to the reader what statistical test(s) were used. As the cross-tabulation is mentioned in the first sentence one may suppose that chi-square test was used to check the association between bullying and gender and age. Please clarify what statistical procedure was used to evaluate relationships described in this fragment.

AUTHORS RESPONSE (Page 4, line 137): We now add the type of test in the following sentence: ‘In the present data, chi-square tests in cross-tabulations showed that gender and age were not significantly associated with bullying (p = 0.484; p = 0.501). Bullied workers had a lower occupational level (p = 0.001) and higher physical demands at work (p = <0.000), and were more likely to be smokers (p = 0.010) (Tables not shown).’

2.4. Finally, a remark about citing (references). The journal uses a different way of giving references than this used in the manuscript, i.e. with consecutive numbers and not authors’ names and year of publication. According to IJERPH Instructions for Authors: “References must be numbered in order of appearance in the text (including table captions and figure legends) and listed individually at the end of the manuscript”.

AUTHORS RESPONSE: We apologise for having used a totally wrong format. It is now corrected throughout the manuscript.